# Von Willebrand Factor Antigen, Biomarkers of Inflammation, and Microvascular Flap Thrombosis in Reconstructive Surgery

**DOI:** 10.3390/jcm13185411

**Published:** 2024-09-12

**Authors:** Rihards Peteris Rocans, Janis Zarins, Evita Bine, Insana Mahauri, Renars Deksnis, Margarita Citovica, Simona Donina, Indulis Vanags, Sabine Gravelsina, Anda Vilmane, Santa Rasa-Dzelzkaleja, Biruta Mamaja

**Affiliations:** 1Intensive Care Clinic, Riga East University Hospital, Hipokrata Street 2, LV-1079 Riga, Latvia; evitabine@gmail.com; 2Department of Anaesthesia and Intensive Care, Riga Stradiņš University, Dzirciema Street 16, LV-1007 Riga, Latvia; insana.mahauri@rsu.lv (I.M.); indulis.vanags@rsu.lv (I.V.); biruta.mamaja@aslimnica.lv (B.M.); 3Department of Hand and Plastic Surgery, Microsurgery Centre of Latvia, Brivibas Street 410, LV-1024 Riga, Latvia; janis.zarins@mcl.lv; 4Baltic Biomaterials Centre of Excellence, Headquarters at Riga Technical University, Pulka Street 3, LV-1007 Riga, Latvia; 5Surgical Oncology Clinic, Riga East University Hospital, Hipokrata Street 4, LV-1079 Riga, Latvia; renars.deksnis@gmail.com; 6Laboratory Department, Riga East University Hospital, Hipokrata Street 2, LV-1079 Riga, Latvia; margarita.citovica@aslimnica.lv; 7Institute of Microbiology and Virology, Riga Stradins University, Ratsupites Street 5, LV-1067 Riga, Latvia; donsimon@inbox.lv (S.D.); sabine.gravelsina@rsu.lv (S.G.); anda.vilmane@rsu.lv (A.V.); santa.rasa-dzelzkaleja@rsu.lv (S.R.-D.); 8Outpatient Department, Riga East University Hospital, Hipokrata Street 4, LV-1079 Riga, Latvia

**Keywords:** von Willebrand factor antigen, neutrophil-to-lymphocyte ratio, flap loss, microvascular flap complications, microvascular flap thrombosis, reconstructive surgery

## Abstract

**Background**: Microvascular flap surgery has become a routine option for defect correction. The role of von Willebrand factor antigen (VWF:Ag) in the pathophysiology of flap complications is not fully understood. We aim to investigate the predictive value of VWF:Ag for microvascular flap complications and explore the relationship between chronic inflammation and VWF:Ag. **Methods**: This prospective cohort study included 88 adult patients undergoing elective microvascular flap surgery. Preoperative blood draws were collected on the day of surgery before initiation of crystalloids. The plasma concentration of VWF:Ag as well as albumin, neutrophil-to-lymphocyte ratio (NLR), interleukin-6, and fibrinogen were determined. **Results**: The overall complication rate was 27.3%, and true flap loss occurred in 11.4%. VWF:Ag levels were higher in true flap loss when compared to patients without complications (217.94 IU/dL [137.27–298.45] vs. 114.14 [95.67–132.71], *p* = 0.001). Regression analysis revealed the association between VWF:Ag and true flap loss at the cutoff of 163.73 IU/dL (OR 70.22 [10.74–485.28], *p* = 0.043). Increased VWF:Ag concentrations were linked to increases in plasma fibrinogen (*p* < 0.001), C-reactive protein (*p* < 0.001), interleukin-6 (*p* = 0.032), and NLR (*p* = 0.019). **Conclusions**: Preoperative plasma VWF:Ag concentration is linked to biomarkers of inflammation and may be valuable in predicting complications in microvascular flap surgery.

## 1. Introduction

Microvascular flap surgery has become a routine option for the correction of various defects during the past decades. Despite improvements in surgical reconstructive techniques, flap loss still occurs, and the rate of flap loss varies from 6 to 10% [1,2,3]. The success of microvascular flaps depends on a multitude of variables, including technical factors, blood rheology, and coagulogy, as well as patient comorbidities [4]. Many recent studies have outlined the use of preoperative biomarkers to predict complications in microvascular flap surgery [5,6]. There is potential to improve perioperative care, assess the risk of flap loss, and understand the pathophysiology of microvascular flap problems through the use of laboratory biomarkers [5]. In recent studies, different patterns of pathophysiology for distinct complications have been outlined [7]. For example, secondary flap complications have been associated with the risk of malnutrition [8,9] as well as chronic inflammation [8,9,10]. The main biomarkers of chronic inflammation that have been associated with flap complications are low albumin [9,11], increased C-reactive protein (CRP) [12], and increased neutrophil-to-lymphocyte ratio (NLR) [13]. Given the pathophysiological distinctions between secondary flap complications and flap thrombosis, the exact role of chronic inflammation in the pathophysiology of flap thrombosis is not fully understood. True flap loss has been associated with increased fibrinogen [14,15], von Willebrand factor (VWF) function [15], and VWF antigen [15,16,17]. Elevated levels of VWF antigen have been linked to a variety of thrombotic conditions [15,16,17,18,19], suggesting that it could be used as a thrombosis marker in reconstructive surgery [15,16,17]. Previous studies indicate a complex interaction between VWF and inflammatory biomarkers in the context of thrombotic events [18,19,20]. Elevated VWF antigen levels can be caused by endothelial damage or inflammation [18,19], both of which often occur during surgery [16]. VWF, once released, can bind to platelets and collagen, causing platelet adhesion and aggregation at the injury site [20]. Specifically, for microvascular flap surgery, the study by Handschel et al. [15] and the case report by Rothweiler et al. [16] indicate a significant impact of VWF antigen concentration on true flap loss. Moreover, the relationship between elevated VWF antigen and other risk factors associated with the underlying indications for reconstruction, as well as chronic inflammation, is not fully understood [21]. It is also uncertain how VWF antigen concentrations, minor flap complication, and true flap loss are related to one another [15]. CRP [22] and interleukin-6 (IL-6) [23] levels are also considerably elevated during surgery, indicating an inflammatory response that may influence thrombus formation. Specifically for microvascular flap surgery, Du et al. found elevated VWF antigen and CRP levels postoperatively after flap venous crisis in animal models [17]. The aim of this study is to determine the predictive value of VWF antigen for microvascular flap complications and to investigate the relationship between chronic inflammation and increased VWF antigen in various complication types.

## 2. Materials and Methods

### 2.1. Patient Selection, Perioperative Considerations and Outcome Definitions

This prospective observational cohort study included 88 patients undergoing elective microvascular flap transplantation surgery at Riga East University Hospital from 1 October 2021 to 31 March 2024. The study protocol and the informed consent form were approved by the Science Department of Riga East University hospital (Approval Number Nr.AP/08-08/22/135) and by the Ethics Committee of Riga Stradins University (Approval Number 22-2/399/2021). The study included adult patients undergoing elective microvascular flap transplantation. The study excluded patients with severe chronic liver or kidney diseases, with cardiovascular and autoimmune diseases, patients with pre-existing coagulopathies or any clotting and bleeding disorders, patients with inherited or acquired von Willebrand disease, patients receiving hormonal contraception or estrogen therapy, patients after recent thrombotic or thromboembolic events, patients currently taking anticoagulants or antiplatelet agents, patients with active systemic infections or inflammatory conditions, pregnant patients and patients during the lactation period, and children under the age of 18. Patients with medication-related osteonecrosis of the jaw, osteoradionecrosis of the jaw, and recent radiotherapy were excluded from the study. Patients with missing or incomplete data were also excluded. The type of flap was chosen based on the type of defect, the length of the pedicle, the positioning during surgery, the body mass index (BMI), and the composition of the patient. Preoperative evaluation, general anesthesia, and postoperative care were provided by a team of experienced attending anesthesiologists. The following flap types were used in the study: anterolateral thigh flap, fibular flap, deep inferior epigastric artery perforator flap, radial free forearm flap, gracilis muscle flap, temporal artery flap, serratus anterior flap, and latissimus dorsi flap. The surgical team closely monitored the microvascular flap for the first 5–7 postoperative days. To monitor flap patency, a clinical assessment of flap color, temperature, tissue turgor, and capillary refill were used. Trauma patients who were operated upon within 30 days of injury were denoted early surgery, and patients who were operated upon later than 30 days from injury were denoted late surgery [21]. True flap loss was defined as the impairment of flap blood supply due to anastomosis dysfunction or thrombosis that leads to a complete loss of the transposed flap. Flap hematoma was defined as the presence of a hematoma adjacent to the flap recipient site without interfering with the flap blood supply. Minor flap complications were defined as the presence of flap wound infection, slow or difficult flap wound healing, marginal flap necrosis, or difficult healing at the donor site. All flap complications were defined as the presence of either true flap loss, flap hematoma or minor flap complications. In all cases of true flap loss, urgent surgical re-exploration was performed.

### 2.2. General Patient Data, Sample Collection, and Laboratory Analysis

General patient data collection was performed according to a previously defined protocol. Written and electronic health records were used to obtain information on the demographic characteristics of the patient, the flap type, the indication for surgery, the recipient location, the duration of surgery, the blood type, perioperative care, and surgical outcomes. Blood draws were obtained on the day of surgery immediately upon the first arrival in the operating room prior to the initiation of the first crystalloid infusion. Full blood count analysis was performed using the XN-1500 system (Sysmex Europe SE, Norderstedt, Germany). Albumin concentrations were analyzed using the colorimetric method (Cobas C, Roche/Hitachi, Mannheim, Germany). Bilirubin concentrations were analyzed using the colorimetric method (Cobas C, Roche/Hitachi, Mannheim, Germany). Interleukin-6 concentrations were analyzed by electrochemiluminescence immunoassay (ECLIA) (Cobas e, Roche/Hitachi, Mannheim, Germany). CRP concentrations were analyzed using the method of immunoturbidimetry (Cobas C, Roche/Hitachi, Mannheim, Germany). The fibrinogen concentrations were analyzed using the CS 5100 system (Sysmex Corporation, Kobe, Japan). The albumin–bilirubin score was calculated using the following formula: albumin–bilirubin score = (log10 bilirubin [µmol/L] × 0.66) + (albumin [g/L] × −0.0852) [24]. The NLR was defined as the proportion of neutrophil count (count/mm^3^) and lymphocyte count (count/mm^3^). All full blood count and clinical chemistry analyses were performed in a clinical laboratory within 8 h after blood draw. All blood samples for VWF antigen analysis were stored within 6 h from blood draw. Prior to storage, blood samples for VWF antigen analysis (collected in citrate tubes) were spun at 3500×/g for 10 min. Plasma samples were then stored at −80 °C until analysis. VWF antigen analysis was performed after a single thaw using the human von Willebrand factor ELISA kit according to the manufacturer’s protocol from Abcam (Cambridge, United Kingdom). All reagents, working standards, and samples were prepared as directed in the product protocol datasheet. The assay was performed at room temperature (20–25 °C) according to the manufacturer’s protocol. The absorbance was read on a Varioskan Lux microplate reader (Thermo Fisher Scientific, Waltham, MA, USA) at a wavelength of 450 nm immediately after the stop solution was added. The obtained readings were grouped for further statistical analysis.

### 2.3. Statistical Analysis

Statistical analysis was performed using SPSS Statistics for Windows, Version 26.0. (IBM Corp. Armonk, NY, USA) and data visualization. GraphPad Prism version 5.03 (GraphPad Software Inc., Palo Alto, CA, USA) was used to present the data graphically. The normality of all variables was tested with visual inspection of the Q-Q plot. The Kolmogorov–Smirnov test was also used to assess whether the datasets conformed to a normal distribution. The Chi-square test and Fisher’s exact test were applied to nominal variable sets. Differences in data distribution between the two groups were evaluated using the Mann–Whitney U test for non-parametric datasets and the two-sample *t*-test for datasets conforming to the normal distribution. Spearman’s rho was used to evaluate correlations between two non-parametric variables. Data sets for Il-6, CRP, fibrinogen, and NLRwere divided into four quartiles. The data distribution differences in VWF antigen were further compared between quartiles using the Kruskal–Wallis H test. Youden’s index (YI) and the concordance probability method (CZ) were used to define optimal cut-off values [25]. Binary logistic regression models were used to obtain odds ratios for specific variables. Continuous variables that conform to normal distribution were presented as a mean and CI95. Statistical significance was assumed if two-tailed *p* < 0.05.

## 3. Results

In total, 88 patients were included, 43 (48.9%) men and 45 (51.1%) women. The mean age was 57.9 years (95% CI95 54.8–61.0). The overall rate of complication was 27.3% (*n* = 24). True flap loss with vascular compromise occurred in 11.4% (*n* = 10), with 4 of these cases being late flap loss (>72 h). Minor flap complications occurred in 10 cases (11.4%), and flap hematoma occurred in 4 (4.5%) cases. All the cases of early true flap loss underwent urgent anastomosis revision. Three cases of late flap loss underwent repeated microvascular flap surgery, and one case underwent necrectomy and reconstruction with a rotated local flap. One patient received intraoperative hemotransfusion, while three patients received hemotransfusion during the early postoperative period. There was no significant relationship between the presence of hemotransfusion and VWF antigen levels.

As depicted in Table 1, there were no significant distinctions in age or gender distribution between the patients with flap complications and the patients without complications. No significant differences in the rates of any flap complications were found between different areas of reconstruction, indications for surgery, or the flap type used.

Trauma was the indication for surgery in nine patients, and the mean time from trauma to surgery was 35.8 days. There was no statistically significant link between time from trauma and VWF antigen concentration (r = −0.67, *p* = 0.865). Regarding time from trauma, no significant differences in the rates of any flap complications were found between early surgery and late surgery groups (*p* = 0.571).

The mean duration of surgery was 6.41 [5.77–7.04] hours. There were no significant differences in the duration of surgery between patients with flap complications and patients without complications (6.68 [5.31–8.06] vs. 6.35 [5.61–7.09], *p* = 0.135). Patients with any flap complications had a significantly lower plasma lymphocyte count (*p* = 0.020); a lower mean plasma albumin (*p* = 0.049); a higher platelet count (*p* = 0.021); and a higher VWF antigen (*p* = 0.014). There was no link between albumin–bilirubin score and VWF antigen concentration (r = 0.89, *p* = 0.430). There were no statistically significant differences in mean bilirubin concentration and albumin–bilirubin score between patients with flap complications and patients without complications (−2.77 [−2.87–−2.67] vs. −2.80 [−3.02–−2.56], *p* = 0.551).

When comparing different surgical indications, trauma had the highest preoperative VWF antigen, followed by oncology, and patients with defects had the lowest VWF antigen (138.62 [107.48–150.62] vs. 129.03 [107.52–150.63] vs. 89.92 [68.09–118.14], *p* = 0.029).

As seen in Figure 1, VWF antigen concentrations were positively linked to preoperative plasma fibrinogen (*p* < 0.001), plasma CRP (*p* < 0.001), plasma IL-6 (*p* = 0.032), and NLR (*p* = 0.019).

VWF antigen levels were higher in true flap loss when compared to patients without complications (217.94 [137.27–298.45] vs. 114.14 [95.67–132.71], *p* = 0.001). Preoperative NLR was the highest in the patients with subsequent secondary flap complications when compared to patients without flap complications (4.36 [2.60–6.13] vs. 2.32 [2.03–2.32], *p* = 0.024). Fibrinogen levels were higher in the patients with subsequent true flap loss compared to patients without complications (5.00 [4.29–5.70] vs. 3.46 [3.18–3.73], *p* < 0.001) (Figure 2). 

After adjustment for fibrinogen and the presence of other types of flaps, logistic regression analysis revealed the association between VWF antigen and true flap loss at the selected cutoff level of 163.73 IU/dL (OR 70.22 [10.74–485.28], *p* = 0.043). 

## 4. Discussion

The main findings of the present study were that biomarkers of inflammation and VWF antigen levels are linked to complications in microvascular flap surgery. The central finding is that elevated preoperative VWF antigen concentrations increase the odds of true flap loss. Flap complications were also found to be related to decreased lymphocyte count, decreased plasma albumin, increased platelet count, increased plasma fibrinogen, and increased NLR. Increases in CRP, fibrinogen, IL-6, and NLR were also directly linked to increases in VWF antigen concentration.

Recent studies have demonstrated that certain aspects of overall health of patients, such as malnutrition [7] and inflammation [10,26], are associated with flap complications. Inflammatory reactions of various kinds, such as infection, malignancy, and autoimmune diseases, also generally show increased fibrinogen activity and increased VWF antigen concentrations [27,28,29,30,31]. VWF is a large glycoprotein produced by endotheliocytes and megakaryocytes [32], and it plays a crucial role in normal hemostasis by enabling platelet plug formation at the sites of vascular injury [33]. VWF mediates platelet adhesion by binding to the collagen and platelet receptors [34]. VWF is highly shear-sensitive and is therefore known to promote platelet aggregation at post-stenotic sites that have a negative shear rate gradient [35]. Studies on atherosclerotic plaques reveal that interactions between the plaque geometries, local endothelial VWF release, and plasma levels of VWF antigen can promote thrombosis [36]. The concept of VWF elongation and subsequent contribution to platelet aggregation at post-stenotic sites may be applicable to the anastomosis site in microvascular flap surgery [15,16,17].

Although specific mechanisms govern VWF-mediated platelet aggregation and subsequent thrombosis, the overall concentration of VWF antigen in plasma has been shown to predict thrombotic events. Specifically for microvascular flap surgery, Handschel et al. found higher levels of plasma VWF antigen concentration in microvascular flap surgery patients with thrombosis of the anastomosed vein [15], which coincides with our findings.

Among the multitude of factors influencing VWF antigen concentrations, chronic inflammation seems to be very prevalent in the microvascular flap surgery patient population [10,13]. Hypercoagulable states may arise due to the primary indication for reconstructive surgery [21], and therefore, all indications for surgery may not imply the same risk of flap loss. The two main indications for reconstruction that can increase the risk of thrombosis are oncology [37,38] and trauma [39,40]. Both aforementioned indications have been associated with increased plasma VWF antigen concentrations [18,27,41]. This coincides with our findings, as our data revealed that trauma has the highest VWF antigen, followed by oncology. Our data revealed that patients with uncomplicated defects as an indication for reconstruction had the lowest VWF antigen concentrations, although all three indication groups had mean values comparable to the previously described normal ranges [42].

Increases in plasma VWF antigen concentrations have previously been associated with indicators of inflammatory states such as increased Il-6 [42], decreased albumin [43], increased CRP [42], increased platelet count [44] and chronic liver disease [45]. Our data revealed that VWF antigen concentration is not linked to albumin–bilirubin score, although no patients in our cohort had chronic liver disease, and the mean albumin–bilirubin score of our patient cohort was expectedly correlated with low risk of hepatic decompensation [46]. We found that microvascular flap complications are related to all included biomarkers of inflammation, except for CRP and IL-6. Furthermore, we found that increased NLR and decreased albumin are linked to secondary flap complications, and increased fibrinogen is linked to true flap loss. These findings further support the notion that true flap loss and secondary flap complications have different patterns of pathophysiology [7]. Chronic inflammation may cause secondary flap complications through impaired healing, much like in any other surgical population [47,48]. However, our data revealed that the presence of chronic inflammation is also associated with an increase in fibrinogen and VWF antigen concentrations. Both fibrinogen and VWF antigen have been shown to increase the risk of true flap loss [15], which coincides with our findings.

This study has several limitations. First, the single-center design prevents generalizability across populations and may have institutional bias. Second, while our sample size of 88 patients is sufficient to provide statistically significant results, due to the limited sample size, multiple risk factors described in the previous literature could not be accounted for in the regression models. Our analysis of the link between true flap loss and VWF antigen concentration did not include the distinction between arterial and venous thrombosis of the anastomosis, which has been reviewed in previous studies [15,16]. Furthermore, collecting blood samples before surgery does not account for changes in biomarker levels before and after surgery. Continuous monitoring of VWF antigen and inflammatory markers during the perioperative phase would provide a more complete understanding of flap complication pathophysiology. The exclusion criteria, which excluded patients with various comorbidities and conditions, may have created selection bias. This could have had an impact on the external validity of the study because excluded populations may have different levels of VWF antigen and inflammatory biomarkers. Specifically, patients with osteoradionecrosis of the jaw were excluded to avoid confounding factors, although inclusion of these cases would have provided valuable insight into the pathophysiology of flap failure after radiotherapy [49]. Our study design and cohort size did not allow for evaluating the predictive power and cost efficiency of VWF antigen when compared with other commonly proposed biomarkers. Larger, multi-center investigations are required to validate the applicability of preoperative VWF antigen analysis to a wider population and to propose specific recommendations for treatment strategies.

## 5. Conclusions

The preoperative level of VWF antigen is associated with true flap loss, while markers of chronic inflammation are linked to both secondary flap complications and increased plasma VWF antigen. Assessment of the preoperative plasma VWF antigen concentration may be valuable for predicting complications in reconstructive surgery. Understanding the pathophysiological link between chronic inflammation, preoperative plasma VWF antigen, and true flap loss may improve decision making in the perioperative care of microvascular flap surgery patients.

## Figures and Tables

**Figure 1 jcm-13-05411-f001:**
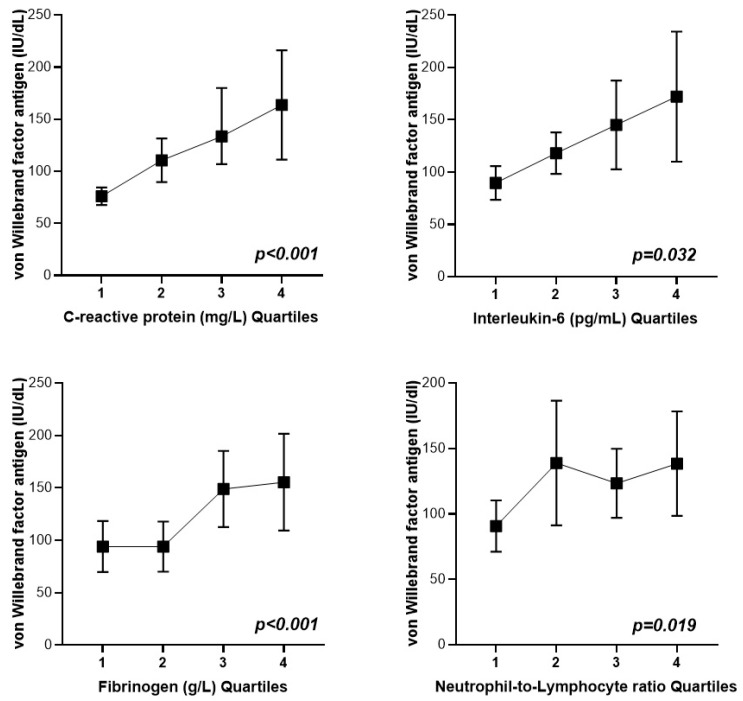
Association between preoperative von Willebrand factor antigen and different preoperative markers of inflammation in microvascular flap surgery patients. Quartiles of C-reactive protein (mg/L): quartile 1 < 0.83; quartile 2 0.84–2.37; quartile 3 2.38–7.81; quartile 4 > 7.82. Quartiles of interleukin-6 (pg/mL): quartile 1 < 5.11; quartile 2 5.12–8.08; quartile 3 8.09–14.80; quartile 4 > 14.81. Quartiles of fibrinogen (g/L): quartile 1 < 2.81; quartile 2 2.82–3.46; quartile 3 3.47–4.27; quartile 4 > 4.28. Quartiles of neutrophil-to-lymphocyte ratio: quartile 1 < 1.47; quartile 2 1.48–1.93; quartile 3 1.94–2.77; quartile 4 > 2.78. Data are presented as mean (CI95).

**Figure 2 jcm-13-05411-f002:**
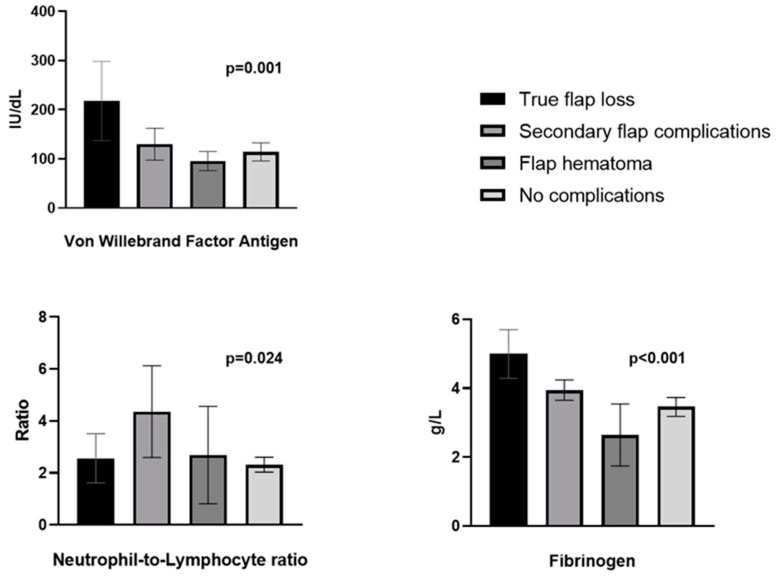
Association between preoperative von Willebrand factor antigen, neutrophil-to-lymphocyte ratio, fibrinogen, and different flap complication types and Kruskal–Wallis H test distribution comparisons of preoperative von Willebrand factor antigen, neutrophil-to lymphocyte ratio and fibrinogen levels in different surgical outcome groups.

**Table 1 jcm-13-05411-t001:** Demographic characteristics, perioperative considerations, comorbidities and laboratory results; data are presented as mean (95% CI) or count (percentage); abbreviations—ENT (ear, nose, and throat surgery); ALT (anterolateral thigh flap); DIEP (deep inferior epigastric artery perforator flap); NLR (neutrophil-to-lymphocyte ratio); CRP (C-reactive protein); VWF (von Willebrand factor).

Patient Group	Overall *n* = 88	No Complications *n* = 64	Any Flap Complications *n* = 24	*p*-Value
**Demographic data**				
Mean age, years	57.9 (54.8–61.0)	57.1 (54.4–59.8)	50.6 (47.7–53.1)	0.204
Women, *n* (%)	45 (51.1%)	31 (48.4%)	10 (41.67%)	0.271
**Location**				
Extremity, *n* (%)	18 (20.5%)	12 (18.8%)	6 (25.0%)	0.603
ENT, *n* (%)	40 (45.5%)	30 (46.9%)	10 (41.7%)	0.787
Head and neck, *n* (%)	17 (19.3%)	12 (18.8%)	5 (20.8%)	0.857
Breast, *n* (%)	13 (14.8%)	10 (15.6%)	3 (12.5%)	0.750
**Flap type**				
ALT, (%)	40 (45.5%)	33 (51.6%)	7 (29.2%)	0.232
Fibular flap, (%)	9 (10.2%)	5 (7.8%)	4 (16.7%)	0.279
DIEP, *n* (%)	10 (11.4%)	8 (12.5%)	2 (8.3%)	0.622
Radial artery flap, *n* (%)	7 (8.0%)	5 (7.8%)	2 (8.3%)	0.941
Other, *n* (%)	22 (25.0%)	13 (20.3%)	9 (37.5%)	0.212
**Indication**				
Trauma, *n* (%)	9 (10.2%)	7 (10.9%)	2 (8.3%)	0.106
Oncology, *n* (%)	55 (62.5%)	42 (47.7%)	13 (54.2%)	0.629
Defect, *n* (%)	16 (18.2%)	10 (11.4%)	6 (25.0%)	0.406
**Blood type**				
Blood type, O, *n* (%)	35 (39.8%)	26 (40.6%)	9 (37.5%)	0.539
**Laboratory values**				
Leukocyte count 10^9^/L	6.24 (5.76–6.73)	6.16 (5.56–6.75)	6.50 (5.62–7.38)	0.309
Lymphocyte count 10^9^/L	1.74 (1.53–1.94)	1.83 (1.58–2.09)	1.41 (1.14–1.68)	0.020
Neutrophil count 10^9^/L	3.66 (3.23–4.09)	3.54 (3.02–4.07)	4.06 (3.28–4.84)	0.109
NLR	2.57 (2.11–3.02)	2.32 (1.85–2.80)	3.4 (2.16–4.64)	0.006
Monocyte count 10^9^/L	0.56 (0.51–0.61)	0.56 (0.50–0.61)	0.56 (0.46–0.66)	0.839
Red blood cell count 10^9^/L	4.08 (3.97–4.19)	4.18 (4.03–4.32)	4.27 (4.06–4.49)	0.625
Platelet count 10^9^/L	246.75 (229.25–264.24)	232.98 (216.00–249.96)	288.37 (238.35–338.40)	0.021
Hemoglobin g/dL	12.58 (12.22–12.94)	12.45 (12.03–12.87)	12.86 (12.13–13.61)	0.208
Mean total plasma protein g/L	64.72 (63.40–66.05)	64.84 (63.27–66.40)	64.08 (61.32–66.83)	0.543
Mean plasma albumin, g/L	38.90 (38.12–39.69)	39.21 (38.25–40.16)	36.58 (34.33–38.84)	0.049
Mean plasma bilirubin, mg/dl	0.47 (0.42–0.53)	0.48 (0.41–0.55)	0.43 (0.31–0.55)	0.447
Mean albumin–bilirubin score	−2.78 (−2.87–−2.69)	−2.80 (−3.02–−2.56)	−2.77 (−2.87–−2.67)	0.551
CRP, mg/L	7.10 (4.67–9.54)	6.76 (3.67–9.86)	7.93 (4.32–11.54)	0.058
Mean plasma fibrinogen, g/L	3.59 (3.35–3.84)	3.52 (3.24–3.81)	3.82 (3.30–4.35)	0.505
Interleukin-6, pg/mL	13.9 (10.09–16.29)	13.03 (9.33–16.7)	13.73 (7.26–20.20)	0.893
VWF antigen, IU/dL	129.61 (111.93–147.27)	120.44 (99.43–141.36)	157.59 (123.62–191.61)	0.014

## Data Availability

The datasets used and analyzed during the current study are available from the corresponding author upon reasonable request. The corresponding author will ensure that individual privacy is not compromised during the transfer of datasets.

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
