# Peer review of "Von Willebrand Factor Antigen, Biomarkers of Inflammation, and Microvascular Flap Thrombosis in Reconstructive Surgery"

_jcm, 2024, doi:10.3390/jcm13185411_

Round 1

Reviewer 1 Report

Comments and Suggestions for Authors

This prospective work does merit to be published, but there is a main concern that needs to be explained. Do you mean that we shall do en evaluation in all our patients of VWF antigen before surgery?. 

if levels are high, what is your recommendation?.

I observed that te number of complications is quite high, and also the percentage of flap loss. It could be another potential factor to alter the final results.

Author Response

Dear Reviewer 1,

Thank you for taking the time to review and improve our draft. We highly value your kind suggestions, ideas, and valuable insights.

The following is a detailed transcript of our feedback to Your kind suggestions:

This prospective work does merit to be published, but there is a main concern that needs to be explained. Do you mean that we shall do en evaluation in all our patients of VWF antigen before surgery?.

Thank you for the valid inquiry. Our study links VWF antigen to flap complications, however since our study design does not allow for accurate comparisons of predictive power and cost efficiency, we cannot propose such a recommendation. We have made sure to expand on this limitation in the Discussion (Line 303-308).

We believe that such routine VWF antigen analysis for all microvascular surgery patients currently may be too costly for our medical care facilities. However, the cost and availability of laboratory technology is usually subject to change and the situation could be different in the future.

if levels are high, what is your recommendation?.

Thank you for the interesting inquiry. Our study links increases of VWF antigen concentration to markers of chronic inflammation. However, since our study design is observational in nature, none of the cases had their treatment or surgical strategies changed in response to the VWF antigen concentration. Therefore, we cannot propose such a recommendation in the paper itself. We have made sure that this limitation is included in the Discussion (Line 305-308).

However, the clinicians in our Authors team would like to mention that changing the timing of surgery and treating modifiable underlying chronic inflammation prior to surgery in regard to VWF antigen and other biomarkers may be a strategy that could be tested in future studies.

I observed that te number of complications is quite high, and also the percentage of flap loss. It could be another potential factor to alter the final results.

Thank you for the valuable question. The overall complication rate was very similar to a recent study by Lese et al (2021, doi.org/10.1016/j.bjps.2020.08.126), which we have made sure to include in our Introduction (Line 45-46). Multiple cohort characteristics are potential confounding factors, which we have made sure to further emphasize in the Discussion (Line 288-291). Our study uses a very meticulous definition for any flap complications, which may include more than other studies report and therefore may reflect a higher percentage value. Importantly, all our observed patients with true flap loss received subsequent revision or new flap transposition and had a successful result in the end, as outlined in the Results section (Line 172-175).

We express our sincere gratitude for the effort and expertise that you contributed towards improving our article!

Kind regards,

The Author team

*Please note that some of the changes in the revised draft are due to suggestions made by other reviewers*

Reviewer 2 Report

Comments and Suggestions for Authors

I consider the topic relevant to the field of reconstructive surgery. However, it does not bring new information, as it rather supports the existing data in the literature.

                I suggest that the authors include the following information in their research: the time from trauma to surgery. Furthermore, it would be interesting to search for a correlation between patients comorbidities (such as chronic liver disease) and the predictive value of VWF antigen for microvascular flap complications. I would, therefore, include two cohorts and compare them. I believe that these changes in the methodology might bring an original touch to the paper.

                The conclusions are consistent with the evidence presented in this study. The authors describe a significantly statistically high level of VWF antigen and fibrinogen in patients with true flap loss compared to those without complications and an association between the increased NLR and the  secondary flap complications, addressing the questions posed in the study.

                Regarding references, I suggest that the authors find more recent data (published within the last 5 years) and also refrain from self-citation (more than 3 times), in order to support their paper.

                Lastly, in table 1 I would split ‘Location, Flap type and Indication’ in separate rows. Also, in Figure 1 I find the interpretation of the displayed statistical data difficult to read.

Author Response

Dear Reviewer 2

Thank you for taking the time to review and improve our draft. We highly value your interest, ideas, and valuable suggestions.

I consider the topic relevant to the field of reconstructive surgery. However, it does not bring new information, as it rather supports the existing data in the literature.

We are pleased to see You found our topic relevant to the field of reconstructive surgery. We have diligently taken Your suggestions into consideration.

The following is a detailed transcript of our feedback to Your queries and suggestions:

I suggest that the authors include the following information in their research: the time from trauma to surgery.

Thank You for the kind suggestion. We have included the analysis of time from trauma to surgery in our Results section (Line 187-191) and the definitions used for early and late surgery are included in the methods section (Line 108-110).

Furthermore, it would be interesting to search for a correlation between patients comorbidities (such as chronic liver disease) and the predictive value of VWF antigen for microvascular flap complications. I would, therefore, include two cohorts and compare them. I believe that these changes in the methodology might bring an original touch to the paper.

Thank You for the kind suggestion. Unfortunately, no patients in our cohort group who were analyzed for preoperative VWF antigen concentration had any diagnosis of chronic liver disease. However, we acknowledge that some patients in our cohort may have had some subclinical liver function impairment. We also agree with the notion that liver function may be an important confounder in our analysis of factors affecting VWF antigen concentration. To satisfy these points and to address Your valuable suggestion, we have retrieved more data and added analysis of hepatic function with bilirubin and albumin-bilirubin score (ALBI) to the Methods (Line 133-135), Results (Line 197-201) and Discussion (Line 272-276) sections.

The conclusions are consistent with the evidence presented in this study. The authors describe a significantly statistically high level of VWF antigen and fibrinogen in patients with true flap loss compared to those without complications and an association between the increased NLR and the  secondary flap complications, addressing the questions posed in the study.

Thank You for the kind remarks!

                Regarding references, I suggest that the authors find more recent data (published within the last 5 years)

Many thanks for the kind suggestion. We have taken this into consideration to the best of our ability and replaced multiple references with more recent data to better support our statements (References: 1.; 12.; 18.; 19.; 20.; 22.; 32.; 34.). However, even upon extensive search, we have left some references unchanged as they are distinctly important for the questions posed in our study and have no recently published data they could be viably replaced with (References: 15.; 17.;23.;25.; 36.; 38.; 39.; 44).

and also refrain from self-citation (more than 3 times), in order to support their paper.

In response to your kind suggestion, we have limited our use of self-citation in the paper to only 3 times (Line 52, Line 239, Line 281).

Lastly, in table 1 I would split ‘Location, Flap type and Indication’ in separate rows.

Thank You for the kind suggestion. We wholeheartedly agree and in Table 1 we have split ‘Location, Flap type and Indication’ in separate rows.

Also, in Figure 1 I find the interpretation of the displayed statistical data difficult to read.

Thank You for the comment. The X axis of Figure 1 shows distinct quartiles to more accurately represent the statistical method used for these specific analyses (Kurskal-Wallis test). For clarity, these quartiles are meticulously described in the legend of Figure 1. Dividing the inflammatory marker variables into distinct quartiles has advantages in terms of reproducibility, data transparency and conciseness. We would like to support our choice of leaving this detail unchanged with the fact that a very similar method of data display is used in the current largest population study on VWF antigen and inflammatory markers by Möller et al (2020, doi.org/10.1016/j.cyto.2020.155265).

We express our sincere gratitude for the effort and expertise that you contributed towards improving our article!

Kind regards,

The Author team

*Please note that some of the changes in the revised draft are due to suggestions made by other reviewers*

Reviewer 3 Report

Comments and Suggestions for Authors

The Authors investigated the role of Von Willebrand factor antigen, biomarkers of inflammation, and microvascular flap thrombosis in reconstructive surgery. The results are new, the topic interesting to a global audience. The manuscript would benefit from adding, in the Discussion section, some considerations on the role of coagulopathy COVID-19 related in microsurgical procedures. In this regard, the following reference is missing PMID: 33851743.

Comments on the Quality of English Language

Fine

Author Response

Dear Reviewer 3

Thank you for taking the time to review and improve our draft. We highly value your interest, ideas, and valuable suggestions.

The Authors investigated the role of Von Willebrand factor antigen, biomarkers of inflammation, and microvascular flap thrombosis in reconstructive surgery. The results are new, the topic interesting to a global audience.

We are pleased to see You found our ideas novel and interesting to a global audience. We have diligently taken Your kind suggestions into consideration.

The manuscript would benefit from adding, in the Discussion section, some considerations on the role of coagulopathy COVID-19 related in microsurgical procedures. In this regard, the following reference is missing PMID: 33851743.

Thank You for Your kind suggestion! We wholeheartedly agree with the concept that infections and systemic inflammation may promote coagulopathy and true flap loss. We have added this and the relevant reference in the Discussion (Line 239).

We express our sincere gratitude for the effort and expertise that you contributed towards improving our article!

Kind regards,

The Author team

*Please note that some of the changes in the revised draft are due to suggestions made by other reviewers*

Reviewer 4 Report

Comments and Suggestions for Authors

My compliments for the work, well structured and complete

Abstract complete and well written.

Introduction complete and exhaustive.

Materials and methods complete. I recommend evaluating and discussing or excluding patients with previous RT or sarcoradionecrosis.

Results complete and exhaustive

Discussion complete and exhaustive.

Conclusion exhaustive.

I recommend evaluating and discussing or excluding patients with MRONJ or ORNJ. "MRONJ and ORNJ: When a single letter leads to substantial differences." Terenzi V, Della Monaca M, Raponi I, Battisti A, Priore P, Barbera G, Romeo U, Polimeni A, Valentini V. Oral Oncol. 2020 Nov;110:104817.

Minor revisions to clarify point on RT, very interesting and scientifically valid work.

Author Response

Dear Reviewer 4,

Thank you for taking the time to review and improve our draft. We highly value your interest, ideas, and valuable suggestions.

My compliments for the work, well structured and complete

Thank very much You for the kind review! We are very pleased to see such kind remarks.

Abstract complete and well written.

Introduction complete and exhaustive.

Thank You for the kind comments!

Materials and methods complete. I recommend evaluating and discussing or excluding patients with previous RT or sarcoradionecrosis.

Many thanks for the kind suggestion. Patients with recent radiotherapy, MRONJ and ORNJ were excluded from the study. This has been added to the methods section (Line 96-98).

Results complete and exhaustive

Discussion complete and exhaustive.

Conclusion exhaustive.

Thank You for the kind comments!

I recommend evaluating and discussing or excluding patients with MRONJ or ORNJ. "MRONJ and ORNJ: When a single letter leads to substantial differences." Terenzi V, Della Monaca M, Raponi I, Battisti A, Priore P, Barbera G, Romeo U, Polimeni A, Valentini V. Oral Oncol. 2020 Nov;110:104817.

Minor revisions to clarify point on RT, very interesting and scientifically valid work.

Thank You for the kind suggestion. We have added some interesting considerations on radiotherapy/ORNJ and the relevant reference in the Discussion section (Line 301-303).

Thank You for your very kind review!

We express our sincere gratitude for the effort and expertise that you contributed towards improving our article!

Kind regards,

The Author team

*Please note that some of the changes in the revised draft are due to suggestions made by other reviewers*
